# Does knowledge of liver fibrosis affect high-risk drinking behaviour (KLIFAD)? protocol for a feasibility randomised controlled trial

Mohsan Subhani  ,[1,2] Katy A Jones,[3] Kirsty Sprange  ,[4]
Stefan Rennick-Egglestone,[5] Holly Knight  ,[6] Joanne R Morling,[1,2,6]
Doyo G Enki,[7] Andrew Wragg,[2] Stephen D Ryder[1,2]

For numbered affiliations see end of article.

**Correspondence to**
Mohsan Subhani;
mohsan.subhani@nottingham.ac.uk

## ABSTRACT

**Introduction** Heavy drinkers in contact with alcohol services do not routinely have access to testing to establish the severity of potential liver disease. Transient elastography by FibroScan can provide this information. A recent systematic review suggested providing feedback to patients based on markers of liver injury can be an effective way to reduce harmful alcohol intake. This randomised control trial (RCT) aims to establish the feasibility of conducting a larger national trial to test the effectiveness of FibroScan advice and Alcohol Recovery Video Stories (ARVS) in changing high-risk drinking behaviour in community alcohol services common to UK practice.

**Methods and analysis** This feasibility trial consists of three work packages (WP). *WP1*: To draft a standardised script for FibroScan operators to deliver liver disease-specific advice to eligible participants having FibroScan. *WP2*: To create a video library of ARVS for use in the feasibility RCT (WP3). *WP3*: To test the feasibility of the trial design, including the FibroScan script and video stories developed in WP1 and WP2 in a one-to-one individual randomised trial in community alcohol services. Semi-structured interviews will be conducted at 6 months follow-up for qualitative evaluation. Outcomes will be measures of the feasibility of conducting a larger RCT. These outcomes will relate to: participant recruitment and follow-up, intervention delivery, including the use of the Knowledge of LIver Fibrosis Affects Drinking trial FibroScan scripts and videos, clinical outcomes, and the acceptability and experience of the intervention and trial-related procedures. Data analysis will primarily be descriptive to address the feasibility aims of the trial. All proposed analyses will be documented in a Statistical Analysis Plan.

**Ethics and dissemination** This trial received favourable ethical approval from the West of Scotland Research Ethics Service (WoSRES) on 20 January 2021, REC reference: 20/WS/0179. Results will be submitted for publication to a peer-reviewed journal.

**Trial registration number** ISRCTN16922410.

## INTRODUCTION

Alcohol-related liver disease (ARLD) is the most common cause of cirrhosis in the UK,

### Strengths and limitations of this study

► The Knowledge of LIver Fibrosis Affects Drinking (KLIFAD) trial is the first randomised control trial to evaluate the feasibility of using non-invasive liver stiffness measurement and alcohol recovery video stories as a behavioural intervention.

► The mixed-methods design of the KLIFAD trial will enable us to test the acceptability of trial-specific procedures to participants and key alcohol workers.

► The KLIFAD trial will enable a definitive trial to establish the effectiveness of non-invasive screening for liver fibrosis in community alcohol services.

► The primary limitation of the KLIFAD trial is that it is a single-centre trial which could limit the generalisation of findings.

► Due to the nature of the KLIFAD intervention, blinding is not possible.

and mortality from ARLD has risen significantly over the past three decades. It is now the second most common cause of working life years lost in men and fifth in women.[1 2] Europe has one of the highest prevalence of Alcohol Use Disorders (AUD), involving 15% of men and 3.5% of women.[2] Around 25% of the UK population drink above the UK recommended level of 14 units per week, and 10% are harmful drinkers.[3] The UK's total per capita pure alcohol intake for people age ≥15 years is 11.4 L/annum per person, which is twice the global average of 6.4 L/annum per person.[2 3] Approximately 20%–30% of lifelong drinkers develop liver cirrhosis, and the risk is even higher (35%) among harmful drinkers.[4 5]

ARLD causes no symptoms in its earlier stages; indeed, patients are often unaware they have serious physical health problems until they present with the complications of cirrhosis, for example; ascites, jaundice,

encephalopathy, variceal bleed and liver failure, when the opportunity for treatment and recovery of liver health is significantly reduced.[1 5 6] It is estimated that the cost to the UK of alcohol on health is £3.5 billion per year,[3 7] consuming 3.6% of the National Health Service (NHS) annual budget.[8] In England, there were 5698 alcohol-specific deaths in 2018, the alcohol-specific age-standardised death rate was 11.9/100 000 (male=16.4 female=7.6). Nottingham (UK) has one of the highest (total=18.6, male=26.8 and female 10.2) alcohol-specific age-standardised death rate/100 000 in the country.[9] A recent trial from the USA predicted a 75% increase in age-standardised annual mortality and a 65% increase in decompensated cirrhosis due to ARLD over the next two decades.[10]

Systematic reviews of randomised controlled trials (RCTs) have established that delivering brief advice about alcohol to harmful drinkers helps them reduce their alcohol consumption.[11 12] Most studies were conducted in primary care settings where the prevalence of liver disease is likely to be markedly lower than in specialist alcohol treatment services. In alcohol services, where high levels of physical and psychological dependence on alcohol are frequent, National Institute of Clinical Excellence guidelines state adults with high levels of alcohol dependency should be assessed and offered intensive structured community-based interventions (with or without medical therapy) as these provide the best chance of achieving and maintaining abstinence from alcohol.[13] Most clinical services in the UK are based on these principles. Individual programmes vary by locality with many of these services delivered by non-NHS providers. Despite the delivery of brief advice and other alcohol-related interventions in clinical practice for over two decades, mortality and morbidity due to alcohol misuse continues to rise in the UK.[3] There is a pressing need to optimise existing interventions to reduce harmful alcohol intake and examine effective alternative options.

Early diagnosis of liver fibrosis provides an opportunity to intervene and reduce or stop alcohol intake. This is known to be the most effective way of preventing liver disease progression.[14] Transient elastography by FibroScan (Echosens, France) has been used in primary care (General Practice) settings to detect liver disease in populations identified as having liver disease risk (heavy drinkers and those with type 2 diabetes). These studies showed that screening asymptomatic individuals based on risk for liver disease doubles the rates of liver cirrhosis diagnosis in the primary care populations studied.[15 16] Moreover, a recent systematic review suggested providing feedback to patients based on markers of liver injury can be an effective way to reduce harmful alcohol intake.[17] Access to recovery stories can help address mental health problems and support recovery from addiction.[18 19] Peer support from people who have recovered from alcohol misuse is beneficial in modifying high risk drinking behaviour.[20]

This trial aims to investigate the feasibility and acceptability of conducting an RCT in community specialist alcohol services settings run by Nottingham Recovery Network (NRN) and to test the acceptability of trial interventions (FibroScan and Alcohol Recovery Video Stories, ARVS).

### Selection of the term 'alcohol misuse'

We acknowledge the heterogeneity in language used to describe alcohol use, and also the stigma associated with some commonly used terms, which itself can act as barrier to change.[21] Some terms, such as alcohol use Disorder (AUD), are not well understood in the general population. The concept for this feasibility trial was developed in collaboration with Patient and Public Involvement (PPI) groups. After extensive discussion between the study team and PPI groups, we have opted for the term 'alcohol misuse' as a general term to cover excess alcohol intake, harmful alcohol intake, drinking problems, alcohol dependence, and AUD.

We define alcohol misuse as 'weekly alcohol intake ≥14 units, or an Alcohol Use Disorder Identification Test (AUDIT) score of ≥8, or key alcohol worker and/or physician diagnosis or referral from any other services with problem drinking'.

The other definitions relevant to Knowledge of LIver Fibrosis Affects Drinking (KLIFAD) trial are provided in online supplemental material 1 (SP-Definitions).

## METHODS AND ANALYSIS

KLIFAD is a parallel design feasibility RCT. The trial will be conducted in a single centre in the UK, carried out at three community alcohol services in Nottingham (the Wellbeing Hub, Edwin House and the Primary Care Alcohol Clinic run by the NRN) hosted by Framework and NRN and working in partnership with Nottinghamshire NHS Foundation Trust.

The KLIFAD trial consists of three work packages (WP) (figure 1).

### Work package one (WP1)

WP1 aims to design a standardised script framework for FibroScan operators to deliver liver disease-specific advice to participants having FibroScan as part of the feasibility RCT (WP3).

FibroScan, is an ultrasound technology developed by Echosence, France, which non-invasively asseses liver stiffness. A prototype script for FibroScan has been created in consultation with the existing KLIFAD PPI group covering three ranges of FibroScan scores, normal ≤7 Kilopascal (kPa), intermediate fibrosis 8–15 kPa and advance fibrosis ≥15 kPa. online supplemental material 1 The trial flow chart for WP1 is provided in figure 2.

We will organise separate participant and FibroScan operator focus groups to collect feedback on the prototype scripts. The participant focus group will allow us to investigate the key messages to be included in the script

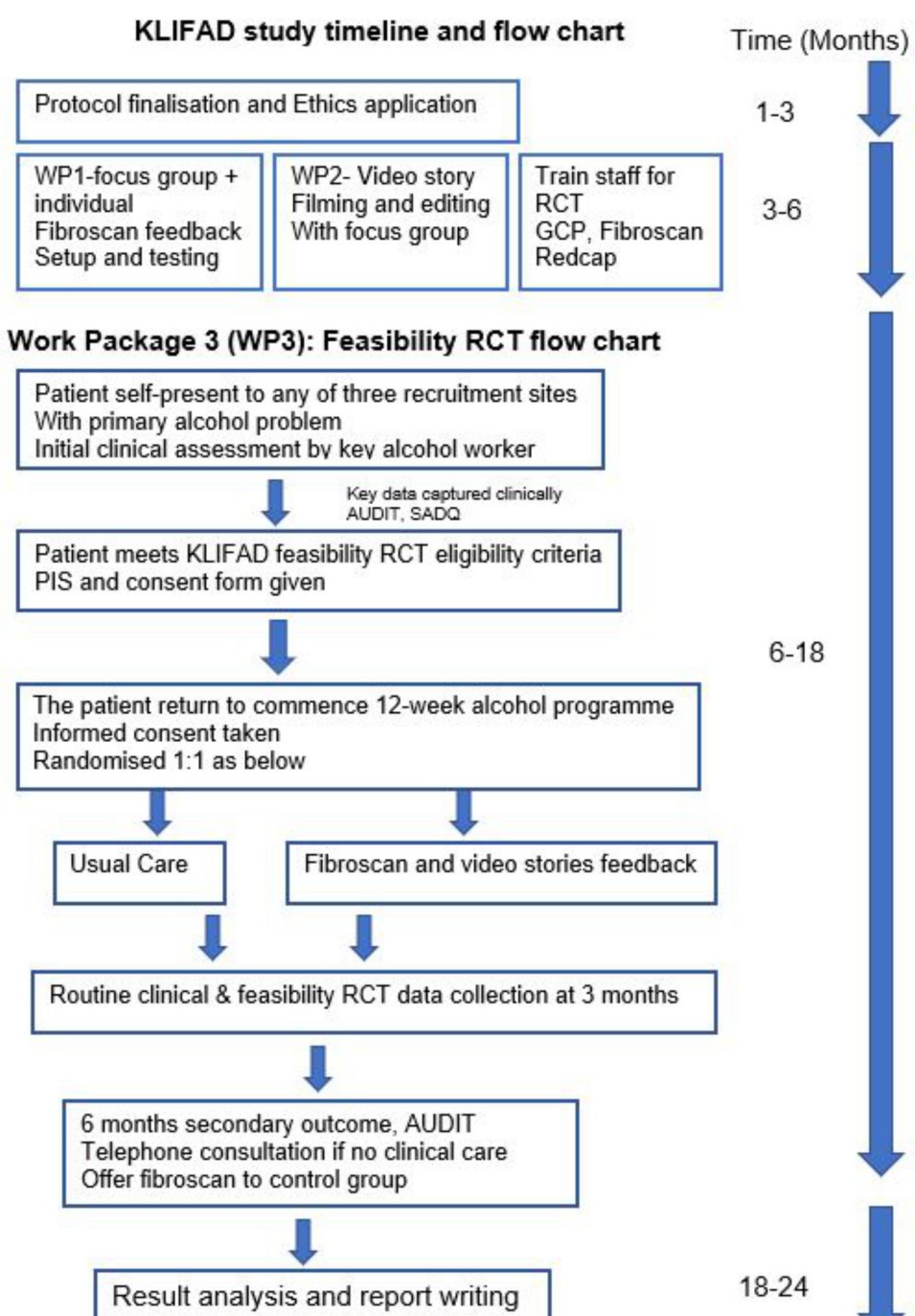

**Figure 1** The KLIFAD trial timeline and flow chart. The trial has three work packages (WP), WP3 is in randomised feasibility trial. AUDIT, Alcohol Use Disorder Identification Test; GCP, good clinical practice; KLIFAD, Knowledge of LIver Fibrosis Affects Drinking; RCT, randomised control trial; SADQ, Severity of Alcohol Dependence Questionnaire.

and feedback, as well as considering how best to present the FibroScan results (eg, graphically, in the text). The FibroScan operator focus group will specifically discuss implementation in clinical practice. In addition, to evaluate the stage of change that each participant has reached, a validated readiness to change model will be piloted.[22]

Following Krueger's (1988) focus group guide, each focus group will include five to eight participants and will last for a maximum of 2 hours.[23] Depending on the latest COVID-19 guidelines, the focus group will be either virtual or face-to-face. A topic guide will be used (SP-Focus Group Guide WP1 V.2.0). We aim to arrange two participant focus groups and one FibroScan operator focus group. The focus groups will be facilitated by two members of the research team. Examples of questions include: (a) If you were a participant in the trial, would the script make sense to you? (b) Are there any parts of the script that you do not understand, and if so, why? (c) What is the best way to present the results of the FibroScan (eg, graphically, in the text)?

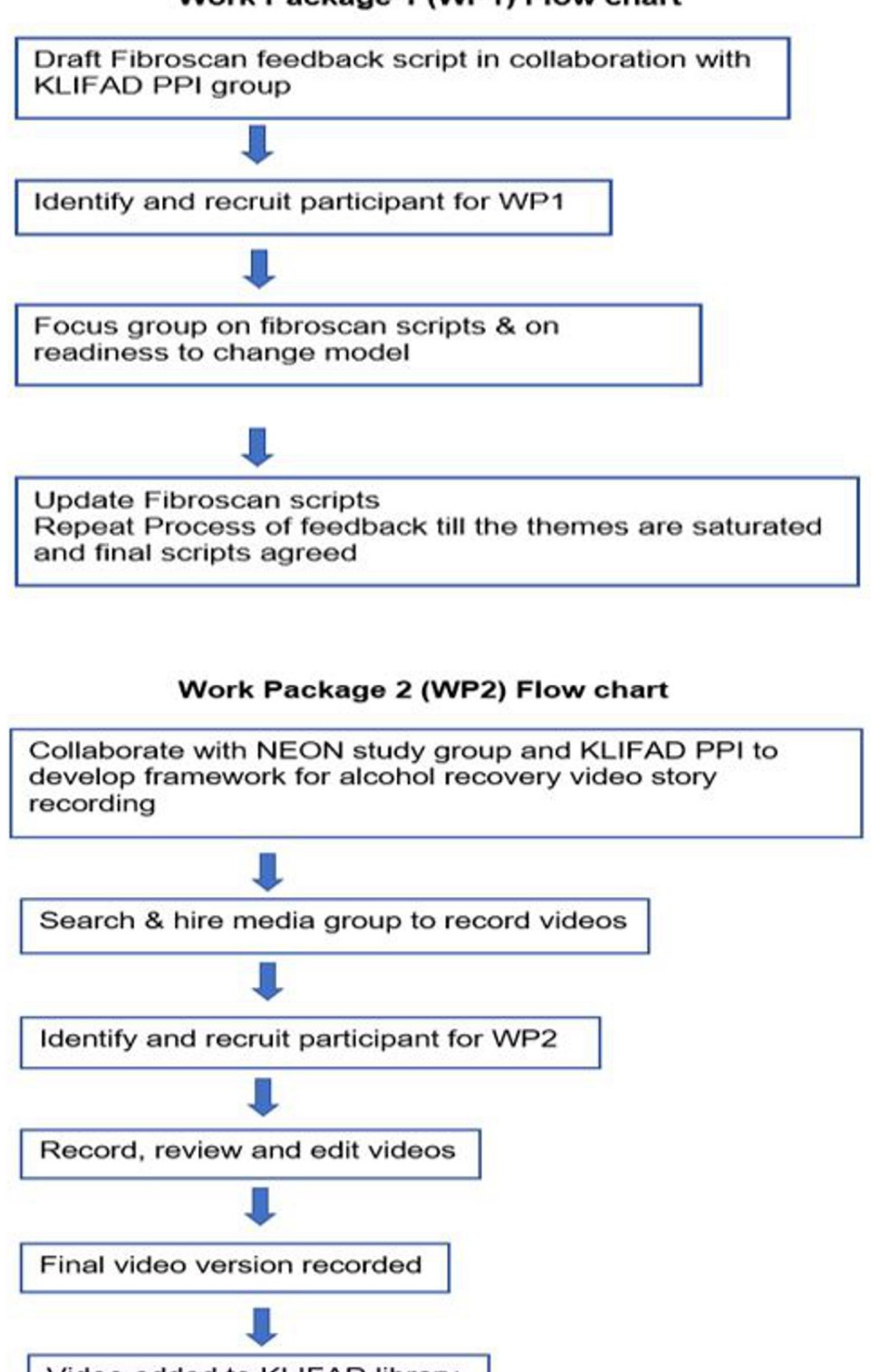

**Figure 2** Flow diagram for; work package one to create FibroScan scripted feedback and work package two to create alcohol recovery videos stories. KLIFAD, Knowledge of LIver Fibrosis Affects Drinking; NEON, narrative experiences online; PPI, patient and public involvement; WP1, work package one.

Eligible participants (table 1) will be identified and recruited through multiple channels. For example, via existing patient forums at all three recruitment settings, the KLIFAD PPI group, by offering information to patients self-presenting to any of the trial treatment settings, snowball methods and via Black, Asian and minority ethnicity/ Framework PPI groups. The focus group meeting will be recorded and transcribed verbatim either by automated

**Table 1** KLIFAD trial eligibility criteria

| Inclusion criteria | Exclusion criteria |
|---|---|
| **Work package one** | |
| A person age ≥18 years | Other primary substance misuse even where alcohol is a factor |
| Primary problem of alcohol misuse‡ | Lacks capacity to confirm consent |
| Had FibroScan in past | |
| **Work package two** | |
| A person age ≥18 years | Lacks capacity to confirm consent |
| Primary problem of alcohol misuse | |
| Had FibroScan in past | |
| A person with lived experience of alcohol problems | |
| A person willing to consent to the recording and public use of their video recording | |
| **Work package three: the randomisation phase** | |
| A person age ≥18 years | Other primary substance misuse even where alcohol is a factor |
| Primary problem of alcohol misuse | Lacks capacity to confirm consent |
| | Referrals from driving offences and student referrals* |
| | Out of area clients at Edwin House† |
| | Participants unable to comply with trial procedures |

*As these individuals are essentially not self-presenting, may have different motivation and have lower overall levels of alcohol use and so are substantially lower risk of having liver disease.
†In whom we cannot obtain follow-up data due to lack of follow-up availability.
‡Alcohol misuse was defined as, weekly alcohol intake ≥14 units, or an AUDIT score of ≥ 8, or key alcohol worker and/or physician diagnosis or referral from any other services with problem drinking.
AUDIT, Alcohol Use Disorder Identification Test; KLIFAD, Knowledge of LIver Fibrosis Affects Drinking.

software or an independent sponsor-approved transcriber. After the first participant focus group, the FibroScan script will be edited considering feedback and a second focus group will then be held to review iterated scripts. The final scripts will be sent via email to participants of focus groups for any final thoughts. We will then organise a FibroScan operator focus group of key alcohol workers working at any of the recruitment settings who are willing to give informed consent, to discuss any specific implementation issues.

After the focus groups, we will collect participant feedback on the change model (SP-Change Model Questionnaire (CMQ) V.1.0) to get an initial sense of the applicability of readiness to change following discussion about the scripts.

### Work package two (WP2)

WP2 aims to create a video library of ARVS from people with a history of alcohol misuse. These ARVS will be used in the feasibility RCT (WP3).

Receiving mental health recovery stories can provide benefits to some people experiencing mental health distress,[18 24 25] and the effectiveness of mental health recovery stories as an intervention to increase quality of life has been examined in a clinical trial.[26] However, equivalent evidence is not available for the impact of ARVS. So that we can explore the impact of stories of recovery from alcohol misuse, in WP2 we will develop a set of recovery stories from participants who have successfully overcome their alcohol misuse. These videos will be peer-reviewed by the KLIFAD PPI group which will include input from Nottingham University Hospitals NHS Trust (NUH) Black, Asian and minority Ethnic PPI group. Based on feedback, the videos will then be edited ready for use in the feasibility RCT (WP3). All edits will be agreed on with the story narrators.

For each narrator, we will follow their preference to create either:
► A recovery story that starts with an open-ended question where narrators have the liberty to tell their story without interruption *or*
► A recovery story in which the participant is asked a set of standard questions.

Drinking history and last FibroScan reading will be included at the start of each video. Eligible participants (table 1) will be recruited through the channels used in WP1. Those who took part in WP1 will also be invited to take part in WP2. A purposive sample based on demographic and liver disease severity of six to nine individuals will be selected.[27] We will arrange a meeting with the KLIFAD PPI group to discuss what makes a video impactful. The outline of WP2 is given in figure 2.

The ARVS will be recorded either at Nottingham Digestive Diseases Biomedical Research Centre (NDDC) Biomedical Research Centre Nottingham University Hospital, the University of Nottingham or the participant's usual place of residence. Each video will be of 2–5 min duration. Videos will be titled based on FibroScan score (low-risk, medium and high-risk score). Videos will be subtitled and depending on the final video format, after the feedback, we envisage adding a photograph of the storyteller and a short-associated text on the title page with informed consent from the participant. The video stories will be brought together in a single-tablet computer-based package from which the participant will be able to choose their most preferred video after receiving a FibroScan score. Collaborative work between a clinician and patient can make a significant impact on

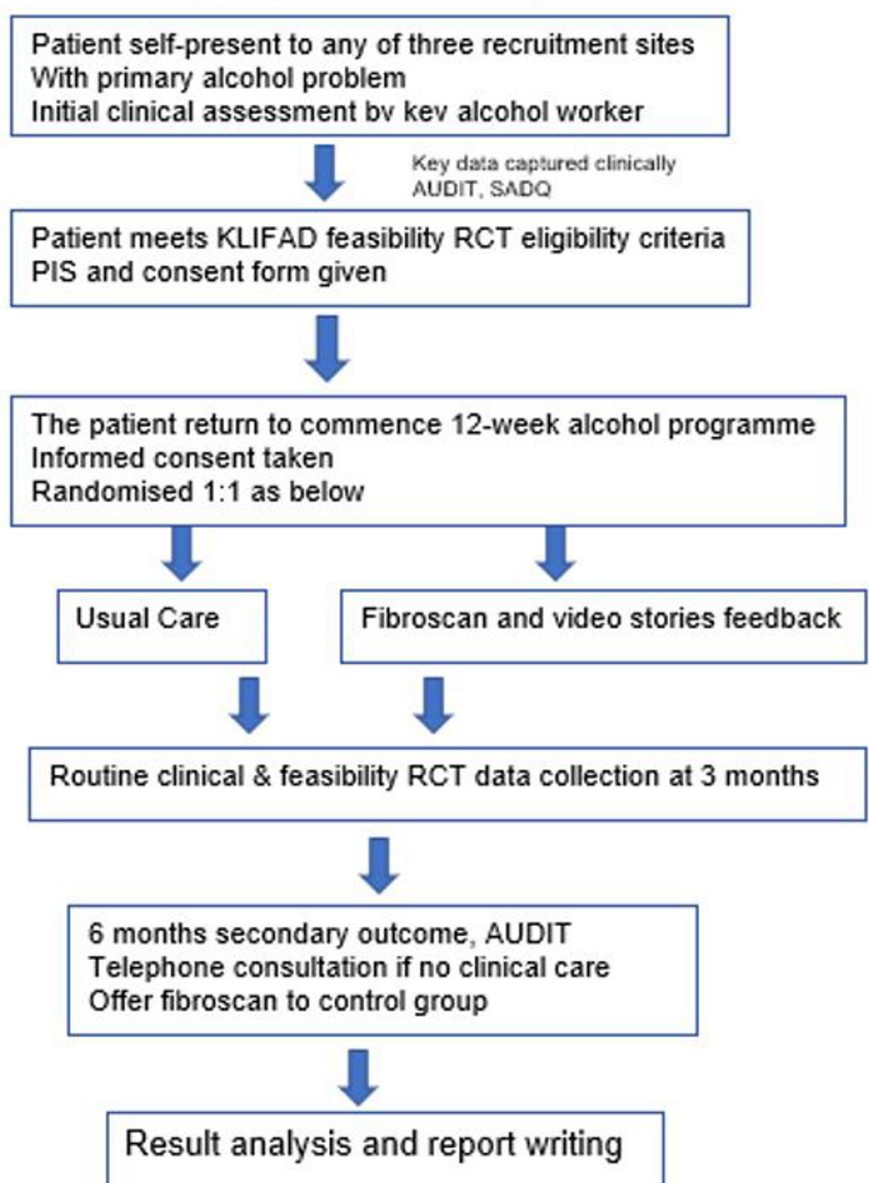

**Work Package 3 (WP3): Feasibility RCT flow chart**

Patient self-present to any of three recruitment sites
With primary alcohol problem
Initial clinical assessment bv kev alcohol worker

Key data captured clinically
AUDIT, SADQ

Patient meets KLIFAD feasibility RCT eligibility criteria
PIS and consent form given

The patient return to commence 12-week alcohol programme
Informed consent taken
Randomised 1:1 as below

Usual Care

Fibroscan and video stories feedback

Routine clinical & feasibility RCT data collection at 3 months

6 months secondary outcome, AUDIT
Telephone consultation if no clinical care
Offer fibroscan to control group

Result analysis and report writing

**Figure 3** Flow diagram for work-package-three, the randomised control trial flow chart. Work package three is feasibility randomised control trial. AUDIT, Alcohol Use Disorder Identification Test; GCP, good clinical practice; KLIFAD, Knowledge of LIver Fibrosis Affects Drinking; RCT, randomised control trial; SADQ, Severity of Alcohol Dependence Questionnaire.

the recovery process[28] and hence in some videos, and with consent by narrators, we will include sections of a video narrated by a clinician the narrator has worked with.

All video stories recorded as part of the KLIFAD trial will have peer review by the trial team and KLIFAD/ Black, Asian and ethnic minority PPI groups. The videos will be shown in the same format that they would be used in WP3.

**Work package 3 (WP3): feasibility RCT**

A feasibility RCT of parallel groups (one-to-one) will compare usual care (assessment and entry into an alcohol reduction programme which does not include information on liver disease severity) to usual care plus feedback

from the FibroScan and ARVS. The eligibility for WP3 is provided in table 1 and the attached flow chart (figure 3).

**Objectives**

Bowen *et al*'s guide for feasibility studies was used to decide objectives.[29]

1. *Test*: the intervention (FibroScan plus feedback and ARVS) in a feasibility RCT.
2. *Acceptability*: of feasibility RCT-related procedures and interventions among patients and healthcare workers.
3. *Feasibility outcomes*: to establish recruitment rate, consent rate, dropout rate and completion rate for accurate sample size calculation for future large-scale RCT.

4. *Refine*: the eligibility and randomisation criteria for a future large-scale RCT.

5. *Implementation and practicality*: to assess the ability of community alcohol services to deliver the intervention, and training and support needs for community alcohol services keyworkers for delivering the intervention.

6. *Adaptation*: of KLIFAD trial interventions, FibroScan feedback and ARVS format and access as per suggestions from participants and key alcohol workers

7. *Limited efficacy*: to test limited efficacy of KLIFAD interventions

## Intervention group

Participants randomised to the intervention arm will receive a FibroScan, feedback on FibroScan results and watch ARVS immediately after. The ARVS will be made available should a participant wish to watch them later.

## Control group

Participants randomised to the control arm will continue with standard treatment (usual care) provided at the three treatment settings. The participants in this arm will be offered FibroScan at 6 months.

As part of standard treatment, the recruitment settings provide different types of interventions to participants in line with the National Drug Treatment Monitoring System Data set (NDTMS) and Public Health England (PHE) guidelines.[30] Existing treatment programmes can run for up to 12 weeks.

For adult drug and alcohol services, there are three main categories of standard intervention (usual care) delivered by the NRN: (a) Psychological: which includes motivational interventions, family and social network interventions and cognitive and behavioural-based relapse prevention interventions (substance misuse specific). (b) Recovery support: which includes 12-step work and counselling.

(c) Pharmacological: which involves prescribing medication for drug and/or alcohol relapse prevention support. For example, naltrexone, acamprosate, disulfiram as part of alcohol or opioid relapse prevention therapy and Chlordiazepoxide for acute alcohol withdrawal.

Specific treatment programmes are started after an initial assessment and based on the participant's needs. The duration of contact with services varies, most participants stay with services for 12 weeks, some get discharged early and a few stay longer than 6 months.

## Methods
### Sample size

As this is a feasibility trial, a formal sample size calculation for between-group comparisons of a primary outcome is not appropriate. Researchers have previously recommended sample sizes between 24 and 50 to satisfactorily achieve feasibility outcomes.[31–33]

After discussion with community alcohol services data manager and considering variation in number of patients presenting per week, we aim to approach 40 eligible participants per month. Assuming a 50% consent rate, we anticipate randomising 20 participants per month (10 per month per arm) for a recruitment period of 6 months. With an estimated sample size of 120, we will be able to calculate a dropout rate of 80% within a 95% CI of ±7.1%. Assuming a non-differential follow-rate of 80%, this target sample size should provide follow-up outcome data on a minimum of 48 participants in each of the two arms.

### Randomisation

The participants will be individually allocated on a one-to-one ratio using minimisation with a probabilistic element. The minimisation variables will be age, gender, ethnicity and severity of alcohol misuse based on the Severity of Alcohol Dependence Questionnaire (SADQ) score. To minimise selection bias, the randomisation will be externally performed by a data manager from NRN.

### Schedule of visits
#### Baseline

The baseline visit will be on the day when the participant starts standard treatment at any recruitment setting. At this visit, written informed consent will be given by participants and participants will be randomised to the intervention or control group. Participants in both arms will have an initial detailed assessment (SP-NRN assessment form online supplemental material 2) as part of their standard care. This includes the collection of baseline demographic and clinical data (eg, age, gender, ethnicity). Participants randomised to the control arm will continue with usual care, while participants randomised to the intervention arm will have the usual care and FibroScan followed by standardised script feedback with ARVS watched immediately after the FibroScan result.

#### Three months

This visit will be part of usual care, no research specific activity will be carried out. The research data will be extracted from routinely collected data from three treatment settings.

#### Six months

This will be a telephone consultation or in-person appointment by the research team. Participants in the control arm will be offered a FibroScan after completion of outcomes. The 6-month follow-up is specifically to cover those who were lost to follow-up at NRN from the treatment programme.

A detailed schedule of the visits is given in table 2.

### Data collection

At baseline, 3 and 6 months, the following data will be collected (table 2)

► Demographics (including address, email address and contact number).
  This will be archived and kept separate from the main database.
► AUDIT scores.
► SADQ scores.

**Table 2** Work-package-three (feasibility RCT) schedule of visits and variables for data

| Trial activity | Baseline visit | 3* months | 6† months |
|---|---|---|---|
| Control group | | | |
| Date and time | Yes | Yes | Yes |
| Baseline consent | Yes | – | – |
| FibroScan+feedback | – | – | Yes |
| Watching video stories | – | – | Yes |
| Qualitative interview | – | – | Yes |
| Demographics | Yes | – | – |
| AUDIT score | Yes | Yes | Yes |
| SADQ score | Yes | Yes | Yes |
| Self-reported alcohol intake‡ | Yes | Yes | Yes |
| Breath alcohol test | Yes | Yes | Yes |
| Substance misuse other than alcohol | Yes | Yes | Yes |
| Data on feasibility outcomes | Yes | Yes | Yes |
| Intervention group | | | |
| Date and time | Yes | Yes | Yes |
| Baseline consent | Yes | – | – |
| FibroScan+feedback | Yes | – | – |
| Watching video stories | Yes | – | – |
| Qualitative interview | – | – | Yes |
| Demographics | Yes | – | – |
| AUDIT score | Yes | Yes | Yes |
| SADQ score | Yes | Yes | Yes |
| Self-reported alcohol intake | Yes | Yes | Yes |
| Breath alcohol test | Yes | Yes | Yes |
| Substance misuse other than alcohol | Yes | Yes | Yes |
| Data on feasibility outcomes | Yes | Yes | Yes |

*3-months visit: this will be routine visit no trial-specific procedure will be carried out.
†6-months visit: will be a telephone consultation and/or if possible/required in person. The participant in the control group will be offered a FibroScan at 6 months if they attend it will be in-person appointment.
‡Self-reported alcohol intake in gram and units per week.
AUDIT, Alcohol Use Disorder Identification Test; RCT, randomised controlled trial; SADQ, Severity of Alcohol Dependence Questionnaire.

▶ Self-reported alcohol intake (gram and unit per week).
▶ Substance misuse other than alcohol.
▶ Breath alcohol testing where participants are still attending.

Breath alcohol testing is a strength of this trial; most studies have relied on self-reporting of alcohol intake. This means we can correlate breath alcohol readings with self-reported alcohol consumption, providing substantial additional information.
▶ Data on feasibility outcomes (eg, screening rate, recruitment rate retention rate).

All the above measurements are part of routine outcome data collected by all three recruitment settings, apart from the 6-month data collected for those who are no longer in a treatment programme at 6 months. All three services included in this trial record all of the above outcomes as part of the 12-week programme standard data set and report these to commissioners. Follow-up data are obtained at every attendance and includes the above data set and breath alcohol testing.

### Qualitative data
We will conduct one-to-one semi-structured interviews to evaluate participant's experiences of being part of the trial (eg, 'Overall, how do you feel about taking part in the KLIFAD trial?') and any changes they may have made to their lives (eg, 'Do you think the KLIFAD trial changed your use of alcohol in any way?'). The preliminary qualitative interview schedule topic guide is provided in online supplemental material 1 (SP: qualitative interview guide). It will be piloted before use by the PPI group to check structure and wording of questions. A readiness to change model used in WP1 will also be piloted. Focus groups and interviews will be audio-recorded and transcribed by an independent transcriber approved by the sponsor, to enable thematic analysis.

### Health economics
Routine NHS data collected for the standard care 12-week treatment programmes will be used together with resources utilisation derived from the NHS digital linked data to derive healthcare costs and the potential benefits of the intervention.

### Outcomes
The outcomes are designed to assess the feasibility and acceptability of the KLIFAD intervention and research processes to help inform a future large-scale RCT. The following outcomes will be reported:
▶ Recruitment rate.
▶ Retention rate.
▶ Consent rate.
▶ Acceptability of the intervention (FibroScan and ARVS).
▶ The willingness of participants to be randomised to trial arms.
▶ Acceptability of the intervention to patients.
▶ Participant adherence.
▶ Feasibility of outcome measures.
These feasibility outcomes will enable the trial team to:
▶ Determine the best primary endpoint for the future definitive trial.

- ► Provide sample size estimates for the future definitive trial.
- ► Record ARVS which will contribute to the video library used in a later large-scale RCT.

## Statistical and data analysis plan

The analyses of the quantitative data will be in line with Consolidated Standards of Reporting Trials guidelines for pilot and feasibility trials.[34] Sekhon *et al*'s framework for acceptability testing will be used.[35] The primary descriptive analyses will be on an intention-to-treat basis (ie, participants are analysed in the group to which they were originally allocated). Data will be summarised using frequency (%), mean (SD) or median (IQR) depending on the distribution of the data. Summary measures will be presented along with their 95% CIs whenever appropriate. Results of the data analysis will be presented using appropriate tables and graphs.

The trial is not powered to investigate statistical significance between the two arms. As this is a feasibility trial, no subgroup analysis is planned. However, the results of the feasibility variables will be presented by categories of different variables (age, gender, ethnicity severity of alcohol misuse).

Different techniques will be followed to maximise the completeness of data collection (eg, via staff training). The level of missing data will be assessed. This is especially useful for the proposed primary outcome variables. An interim analysis is not planned for this trial, but the progress of the trial will be reported to the oversight committee who can assess any concerns.

Thematic analysis of qualitative data will be conducted following Braun and Clarke's standard methods.[36] Care will be taken to integrate updated guidelines about thematic analysis including a transparent appreciation of researcher reflexivity.[36] If the trial management group feel the analysis requires external validity, a sample of transcripts identified by a random number generator with the codebook will be given to a researcher independent of the trial. This will allow us to calculate the % agreement and Cohen's Kappa value (using criteria by Cohen, 1960).[37] The Consolidated Criteria for Reporting Qualitative Studies will be used to ensure thorough and explicit reporting of qualitative data in reports and manuscripts for publication.[38]

## Ethics and dissemination
### Ethical approval
The trial received favourable ethical approval from the West of Scotland Research Ethics Service (WoSRES) on 20 January 2021, REC reference: 20/WS/0179.

### Informed consent
All participants will provide a written or online (e-consent) informed consent before any research activities are initiated. A patient infomation sheet (PIS) written in plain language will be provided and it will be ensured the participant has understood the trial information and had enough time to make an informed decision. The Site Investigator will be available to answer any questions about trial participation.

### Data handling and record-keeping
In compliance with the ICH/Good Clinical Practice guidelines, regulations and following the Nottinghamshire Healthcare NHS Foundation Trust SoPS, the Chief or local Principal Investigator will maintain all records and documents regarding the conduct of the trial. These will be retained for at least 24 months or for longer if required. If the responsible investigator is no longer able to maintain the trial records, a second person will be nominated to take over this responsibility. The routinely collected clinical data will be treated in the same way as other clinical case records are treated in the NHS following relevant policies developed by Nottinghamshire Healthcare NHS Foundation Trust, the UK Government and the National Institute for Health Research (NIHR).

The Trial Master File and trial documents held by the Chief Investigator on behalf of the Sponsor shall be finally archived at secure archive facilities at the NDDC at Nottingham University Hospital NHS Trust (NUHT). This archive shall include all trial databases and associated meta-data encryption codes.

An index will be created for Case Report Forms and paper trial data before storage. All online and IT-based data will be password protected and access will only be granted to people directly involved in trial and data analysis. All patient identifiable data will be pseudonymised with a trial-specific participant number.

The information will be copied to the research database (REDCAP cloud) run by the NUHT. We will delete any information that identifies participants by the end of the KLIFAD trial (currently expected October 2022). All relevant UK data protection laws will be followed, including the 2018 Data Protection Act.

### Participant safety
There is a risk that being given a normal FibroScan result may provide false reassurance and encourage participants to maintain their current level of harmful drinking or encourage them to drink more. It is also possible that a high reading will generate anxiety. The trial is designed to minimise these risks by providing scripted feedback (WP1) and watching ARVS (WP2).

Cirrhosis diagnosis and FibroScan: It is anticipated that a small number of people will be identified who have previously unknown cirrhosis and so would be at risk of complications of liver disease. This will be mitigated by offering onward referral to out-patient Hepatology for all participants with a FibroScan reading >15 kPa. This will be via contact with the participant's GP and would follow the current NUHT Nottinghamshire adult liver disease stratification pathway for referral.[39] Some risk mitigations will be through the feedback included in this trial which covers cirrhosis.

For WP1 and WP2, we cannot foresee any potential risks except possible emotional distress during participation in a focus group or semi-structured interview. Participants can choose to skip any question that they prefer not to answer. If distress occurs during the WP3 trial visit, we will ask the participant to take a break to recover, or they can choose to terminate the process. We do not expect that the trial will cause any discomfort or pose any disadvantages, however, contact details for the trial team are provided should the participant have any questions before, during or after taking part. We have also provided a list of locally relevant support services at the end of each patient information sheet, which participants can consult.

### Patient and public involvement (PPI)

The trial has a dedicated PPI group and has considerable regular input from the PPI group at every stage of the study.

### Dissemination

The results of the feasibility trial will be submitted for publication to a peer-reviewed journal and presented at relevant conferences. A separate manuscript on the qualitative aspect of the trial will be written as well. This work is part of a PhD for the lead author (MS) who will present and submit data as a PhD thesis to the University of Nottingham. The work will also be made available to trial participants via the NDDC Biomedical Research Unit website.

#### Author affiliations

[1]Nottingham Digestive Diseases Biomedical Research Centre (NDDC), School of Medicine, University of Nottingham, Nottingham, UK
[2]NIHR Nottingham Biomedical Research Centre, Nottingham University Hospitals NHS Trust and the University of Nottingham, Nottingham, UK
[3]Academic Unit of Mental Health and Clinical Neuroscience, School of Medicine, C24 Institute of Mental Health, University of Nottingham, Nottingham, UK
[4]Nottingham Clinical Trials Research Unit, University of Nottingham, Nottingham, UK
[5]School of Health Sciences,Institute of Mental Health, University of Nottingham, Nottingham, UK
[6]Division of Epidemiology and Public Health, University of Nottingham, Nottingham, UK
[7]School of Medicine, University of Nottingham, Nottingham, UK

**Contributors** MS: KLIFAD trial: The project is part of PhD thesis. Trial coordinator and member of trial management group. He has and will contribute to following; research idea, funding application, PPI meetings, trial protocol, IRAS application and ethical approval, FibroScan training, site initiation, work package 1 focus group, work package 2 alcohol recovery story recording, monthly trial management group meeting, monitoring ongoing progress of work package 3, qualitative interview, data synthesis and analysis, report writing, dissemination. Manuscript: written initial draft of the protocol, implemented changes and drafted final version of protocol and manuscript. KAJ: KLIFAD trial: member of trial management group. She is supervising the qualitative component of the trial including conducting and analysing semi-structured interviews. Manuscript: reviewed protocol and manuscript, provided specialist input for qualitative aspects of the protocol and contributed to the final manuscript. KS: KLIFAD trial: member of trial management group. She contributed to following; research idea, funding application, trial protocol, IRAS application, work package 3 initiation, trial management and progress. Manuscript: reviewed protocol and manuscript and contributed to the final manuscript. SR-E: KLIFAD trial: member of trial management group. He is supervising work package 2 including proposal for alcohol recovery stories recording, editing and finalising. Manuscript: Reviewed protocol and manuscript, provided specialist input for work-package-2 of protocol and contributed to final manuscript. HK: KLIFAD trial: member of trial management group. She is contributing to work package 1 including developing FibroScan results feedback scripts and organising focus groups. Manuscript: reviewed protocol and manuscript, provided specialist input for work-package-1 of protocol and contributed to final manuscript. JRM: KLIFAD trial: Member of trial management group. She PhD supervisor for Dr Subhani, supervising trial overall and specifically helping with health economics part of trial. Manuscript: reviewed protocol and manuscript, provided specialist input for health economics section of protocol and contributed to final manuscript. DE: KLIFAD trial: member of trial management group. He is statistical support for the trial. Manuscript: reviewed final manuscript. AW: KLIFAD trial: patient and public involvement coordinator.Manuscript: reviewed final manuscript. SDR: KLIFAD trial: Chief investigator, PhD supervisor for Dr Subhani and member of trial management group. He has contributed to following; research idea, funding application, PPI meetings, trial protocol, IRAS application and ethical approval, overall supervision of all three work packages, data synthesis and analysis, report writing, dissemination. Manuscript: reviewed final manuscript

**Funding** This work was supported by 'National Institute for Health Research (NIHR)' grant number (RfPB NIHR201146). JRM receives salary support from a Medical Research Council Clinician Scientist award (grant number MR/P008348/1).

**Competing interests** None declared.

**Patient consent for publication** Not required.

**Provenance and peer review** Not commissioned; externally peer reviewed.

**Data availability statement** Data are available on reasonable request. The anonymised data that will support the findings of this trial will be available from the corresponding author MS upon reasonable request.

#### ORCID iDs

Mohsan Subhani http://orcid.org/0000-0001-8739-7263
Kirsty Sprange http://orcid.org/0000-0001-6443-7242
Holly Knight http://orcid.org/0000-0002-4602-3238

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
