## [Reviewer comments · BMJ Open]

ARTICLE DETAILS

TITLE (PROVISIONAL)	Does knowledge of liver fibrosis affect high-risk drinking behaviour (KLIFAD)?: Protocol for a feasibility randomised controlled trial
AUTHORS	Subhani, Mohsan; Jones, Katy; Sprange, Kirsty; Rennick-Egglestone, Stefan; Knight, Holly; Morling, Joanne; Enki, Doyo; Wragg, Andrew; Ryder, Stephen

VERSION 1 – REVIEW

REVIEWER	Thiele, Maja Odense Universitetshospital, Department for Gastroenterology and Hepatology
REVIEW RETURNED	02-Sep-2021

GENERAL COMMENTS	I appreciate the opportunity of reading this protocol for a feasibility randomized controlled trial. The subject is highly important and needed, and a large-scale randomized trial would certainly move the field of combined alcohol and liver research forward. Overall, I found that the submitted manuscript really describes three different studies: WP1, -2 and -3. This unfortunately makes for confusing reading. As an example, the aims paragraph lists a full 13 objectives, few of which live up to the SMART criteria. I therefore suggest to focus on just one trial's protocol – for example the RCT, which also seem to be the main focus of the manuscript. And to use the SMART criteria for design of objectives. An added effect of including all three studies in the same manuscript is the length. A more narrow focus on the RCT would highly improve readability. Specific comments and suggestions: • Use the generic name instead of FibroScan. E.g. liver stiffness measurement/ transient elastography by FibroScan (Echosens, France).• Typically, alcohol-related liver diseases is abbreviated ALD, or in some cases ArLD• Typically, percentages are reported as integers when >10%, while decimals are allowed when <10%• Inclusion criteria lack a clear definition of what constitutes 'alcohol misuse'. Also, the term misuse is not seen in neither ICD coding or DSM.• I may not know enough about UK ethical committees, or geography, but how come is the study approved by West of Scotland Research Ethics Service, but will take part in Nottingham?• Abstract: Please make clear that randomization is person-based, not center-based. Can be misunderstood.• Introduction: Recommended levels of alcohol intake differ btw countries. Please specify UK recommendation.
--

	 • Please refrain from phrases like “ascites (building fluid into abdomen)”. That is meant for laymen, while a BMJ Open reader would be expected to know the concept of ascites. • I disagree with the sentence “Moreover, a recent systematic review suggested providing feedback to patients based on markers of liver injury is an effective way to reduce harmful alcohol intake¹⁷”. All but two of the studies included in reference 17 compares brief intervention or another structured psychosocial intervention and biofeedback (mostly GGT, which is not a liver fibrosis marker) versus no intervention or unstructured advice. Therefore, the review cannot be used to support that argument. It is likely the psychosocial intervention that is effective, not the feedback based on biochemical markers. • Minor: Misspelled “United Kingdome”
--	--

REVIEWER	Weerasinghe, Ashini Public Health Ontario, Health Promotion, Chronic Disease & Injury Prevention
REVIEW RETURNED	04-Sep-2021

GENERAL COMMENTS	Page 7 out 62 Line 4-UK should be initially spelled out here and not in line 8 Line 10-What % of the UK population 15+ drink? How does the 11.4 litres compare to the global average? Line 21 “When the opportunity for treatment and recovery of liver health are significantly reduced” Should be combined with previous sentence (sentence should not be on its own) Line 23 “In England, in 2018”-repetitive, consider revising Lines 24-25: Consider revising to “the alcohol-specific age-standardised death rate was 11.9/100,000” Page 9 of 62 Line 8: “Each focus group will include 5-8 participants and will last for a maximum of 2 hours” How were these numbers derived? Line 17: “What is the best way to present the results of the FibroScan?” Should this question include (e.g., graphically, in the text) so that participants can be prompted? Page 11 of 62 Lines 39-41 “Videos will be subtitled and depending on the final video format after the feedback we envisage adding a photograph of the storyteller and a short associated text on the title page.” Will consent be considered here? Page 12 of 62 Lines 42-43: “However, we aim to approach 40 eligible participants per month.” Is this based on previous research? Can they be cited? Page 15 of 62 Line 57: Can you elaborate on how the outcomes will be measured? i.e. will existing scales be used to measure acceptability?
--

	Will the quantitative and qualitative results be triangulated? Also, fibroscan is capitalized in some parts of the manuscript and not elsewhere. Keep them the same for consistency. Reference 10 title should be corrected to the following: Projected prevalence and mortality associated with alcohol-related liver disease in the USA, 2019-40: a modelling study
--	---

VERSION 1 – AUTHOR RESPONSE

Reviewer: 1

Comments to the Author:

I appreciate the opportunity of reading this protocol for a feasibility randomized controlled trial. The subject is highly important and needed, and a large-scale randomized trial would certainly move the field of combined alcohol and liver research forward.

Overall, I found that the submitted manuscript really describes three different studies: WP1, -2 and -3. This unfortunately makes for confusing reading. As an example, the aims paragraph lists a full 13 objectives, few of which live up to the SMART criteria. I therefore suggest to focus on just one trial's protocol – for example the RCT, which also seem to be the main focus of the manuscript. And to use the SMART criteria for design of objectives. An added effect of including all three studies in the same manuscript is the length. A more narrow focus on the RCT would highly improve readability.

Specific comments and suggestions:

Thank you very much for this comment. The three work packages (WP) are interrelated, work-package-three (WP3) can only start once work packages one (WP1) and two (WP2) are completed. As both interventions used in WP3 will be developed in WP1 and WP2. We think for readers to understand the study it important to provide overview of WP1 and WP2. As the reviewer suggested we have moved the list of objectives from the introduction and have provided focused objectives for WP3 only. We hope this improve the readability of the protocol.

- Use the generic name instead of FibroScan. E.g. liver stiffness measurement/ transient elastography by FibroScan (Echosens, France).

Thank you very much for your comment. To help reader understand what a FibroScan is, we have added an explanatory statement in in work package one. The stamen reads as “FibroScan, is an ultrasound technology developed by Echosence, France, which non-invasively asseses liver stiffness”. Moreover in abstract and introduction we have changed it to “Transient elastography by FibroScan”

- Typically, alcohol-related liver diseases is abbreviated ALD, or in some cases ArLD
Thank you very much for your comment. National Health Services (NHS), and National Confidential Enquiry into Patient Outcome and Death (NCEPOD) disease use abbreviation of ARLD for alcohol-related liver disease. Moreover, the British Liver Trust which is source of accessible information to patient on liver disease in the UK use abbreviation of ARLD for alcohol-related liver disease. On the suggestion from our PPI group and to be consistent with current UK patient information resources we have opted ARLD as an abbreviation for alcohol-related liver disease. I have provided the links below;

<https://www.nhs.uk/conditions/alcohol-related-liver-disease-arld/>
<https://www.ncepod.org.uk/2013report1/slides/ARLDpresentation.pdf>

<https://britishlivertrust.org.uk/information-and-support/living-with-a-liver-condition/liver-conditions/alcohol/>

- Typically, percentages are reported as integers when >10%, while decimals are allowed when <10%
Thank you very much for your comment. We have made changes where appropriate as suggested by the reviewer. At places we have kept the referenced data as it is as reported by the World health organisation and Public Health England. We felt a rounding up or down can risk over or underestimation of the data and can affect validity.
- Inclusion criteria lack a clear definition of what constitutes 'alcohol misuse'. Also, the term misuse is not seen in neither ICD coding or DSM.

Thank you very much for your suggestion. We agree with the reviewer; alcohol misuse is a complex concept to explain due to heterogeneity in the way it has been defined in the literature.

The Diagnostic and Statistical Manual of Mental Disorders, 5th edition (DSM-5)- integrated alcohol abuse and alcohol dependence into a single disorder called Alcohol Use Disorder (AUD) with mild, moderate, and severe classification.

The National Institute on Alcohol Abuse and Alcoholism (NIAAA) define alcohol misuse as "alcohol consumption that puts individuals at increased risk for adverse health and social consequences"¹. AUD is a medical diagnosis for problem drinking; the NIAAA define AUD as "a chronic relapsing brain disorder characterized by an impaired ability to stop or control alcohol use despite adverse social, occupational, or health consequences"¹.

The World Health Organization International Classification of Diseases -10 (ICD-10) criteria describe Alcohol dependence as "a cluster of physiological, behavioral, and cognitive phenomena in which the use of a substance or a class of substances takes on a much higher priority for a given individual than other behaviors that once had greater value, and harmful or hazardous as a pattern of psychoactive substance use that is causing damage to health"^{2,3}. ICD-

10 use code F10*- for Alcohol related disorders with multiple sub-diagnostic codes.

The National Institute for Health and Care Excellence (NICE) 2011 guideline used AUD term for harmful drinking (high-risk drinking) and alcohol dependence and recommended to use formal assessment tools (AUDIT, SADQ, LDQ) to assess the nature and severity of alcohol misuse.

The National Health Services (NHS) UK define alcohol misuse as "Alcohol misuse is when you drink in a way that's harmful, or when you're dependent on alcohol. To keep health risks from alcohol to a low level, both men and women are advised not to regularly drink more than 14 units a week". We acknowledge the heterogeneity in language used to describe alcohol use and stigma associated with some of these terms, which itself can act as barrier to change. Some of terms like AUD are not well understood among general population. The original research idea for the current research project was put forward in collaboration with patient and population representative group (PPI). After thoughtful discussion between study and PPI groups, we opted term 'alcohol misuse' to describe excess alcohol intake, harmful alcohol intake, drinking problems, alcohol dependence, and AUD. We have provided following definition of alcohol misuse in Table 1. Alcohol misuse was defined as, "weekly alcohol intake ≥ 14 units, or an AUDIT score of ≥ 8 , or key alcohol worker and/or physician diagnosis, or referral from any other services with problem drinking". We have added statement at end of introduction section.

- I may not know enough about UK ethical committees, or geography, but how come is the study approved by West of Scotland Research Ethics Service, but will take part in Nottingham?

Thank you very much for your comment. There are multiple national research ethics committees in the UK managed by Health Research Authority (HRA), and a centralised Integrated Research Application System (IRAS).

URL: <https://www.hra.nhs.uk/about-us/committees-and-services/res-and-recs/>

URL: <https://www.myresearchproject.org.uk/>

All researchers are advised to submit their research proposal for ethical approval through this system. After submission the IRAS administrative team allocate a national ethics committee most suitable to research design which is often not local to the place of research. In our case, this committee was in Scotland.

- Abstract: Please make clear that randomization is person-based, not center-based. Can be misunderstood.
Thank you very much for your suggestion. We have specified randomisation as one-to-one individual.
- Introduction: Recommended levels of alcohol intake differ btw countries. Please specify UK recommendation.
Thank you very much for your comment. We have now specified the UK specific weekly alcohol limit as suggested.
- Please refrain from phrases like “ascites (building fluid into abdomen)”. That is meant for laymen, while a BMJ Open reader would be expected to know the concept of ascites.
Thank you very much for your comment. The layman explanation of terms was added based on recommendations from our PPI group. We have now amended as per reviewer’s recommendation.
- I disagree with the sentence “Moreover, a recent systematic review suggested providing feedback to patients based on markers of liver injury is an effective way to reduce harmful alcohol intake¹⁷”. All but two of the studies included in reference 17 compares brief intervention or another structured psychosocial intervention and biofeedback (mostly GGT, which is not a liver fibrosis marker) versus no intervention or unstructured advice. Therefore, the review cannot be used to support that argument. It is likely the psychosocial intervention that is effective, not the feedback based on biochemical markers.

Thank you very much for your comment. In the referenced systematic review as no group received biofeedback without brief advice, the author attempted to investigate this by looking at biofeedback + brief advice vs a range of alternatives and found (see Table 2 of the original publication) that the biofeedback + brief advice only had an alcohol consumption change mean difference of -78 g/week (P = 0.16) suggesting the biofeedback does have a role to play and stated the limitation of lack of statistical significance and small number of studies $n = 3^4$. Moreover, the study used GGT as marker of liver injury not fibrosis. As author highlighted the uncertainty, we have updated the sentence as “Moreover, a recent systematic review suggested providing feedback to patients based on markers of liver injury can be an effective way to reduce harmful alcohol intake”

- Minor: Misspelled “United Kingdome”
Thank you very much for your comment. We have corrected the spelling mistake.

Reviewer: 2

Comments to the Author:

Page 7 out 62

- Line 4: UK should be initially spelled out here and not in line 8

Thank you very much for your comment. We have fully spelled UK as suggested by reviewer.

- Line 10-What % of the UK population 15+ drink? How does the 11.4 litres compare to the global average?
Thank you very much for your comment. We have provided the global per capita alcohol intake compared to the UK.

- Line 21 “When the opportunity for treatment and recovery of liver health are significantly reduced” Should be combined with previous sentence (sentence should not be on its own)
Thank you very much for your comment. We have combined the sentences as suggested by the reviewer.
- Line 23 “In England, in 2018”-repetitive, consider revising
Thank you very much for your comment. We have re-ordered the sentence as the reviewer suggested.
- Lines 24-25: Consider revising to “the alcohol-specific age-standardised death rate was 11.9/100,000”
Thank you very much for your comment. We have re-ordered the sentence as the reviewer suggested.

Page 9 of 62

- Line 8: “Each focus group will include 5-8 participants and will last for a maximum of 2 hours” How were these numbers derived?
Thank you very much for your comment. The focus group size was decided based on aim and design of study, and primary research question following Krueger (1988) guide to focus groups. We have provided the reference.
- Line 17: “What is the best way to present the results of the FibroScan?” Should this question include (e.g., graphically, in the text) so that participants can be prompted?
Thank you very much for your comment. We have amended the question as per the reviewer’s suggestion.

Page 11 of 62

- Lines 39-41 “Videos will be subtitled and depending on the final video format after the feedback we envisage adding a photograph of the storyteller and a short associated text on the title page.” Will consent be considered here?
Yes, videos will be recorded, titled, subtitles and label with informed consent from the participant. A statement about consent has been added to this section.

Page 12 of 62

- Lines 42-43: “However, we aim to approach 40 eligible participants per month.” Is this based on previous research? Can they be cited?
Thank you very much for your comment. Yes, this was based on historical recommendation of sample sizes between 24-50 to satisfactorily achieve feasibility outcomes⁵⁻⁷. We have updated the section as per the reviewer’s recommendation and have provided relevant references.

Page 15 of 62

- Line 57: Can you elaborate on how the outcomes will be measured? i.e. will existing scales be used to measure acceptability?
Thank you very much for your comment. The primary descriptive analysis will be on intention to treat basis, we will use the Sekhon et al (2017) framework to test acceptability⁸. As this is a feasibility study, limited efficacy testing will be done for trial interventions.
- Will the quantitative and qualitative results be triangulated?
Thank you very much for your comment. As this is a randomised control trial, it should minimise risk of bias and confounders. We are not planning to triangulate the

results. For qualitative analysis an inductive thematic analysis approach will be adopted following Braun and Clarke's standard methods⁹.

- Also, fibrocan is capitalized in some parts of the manuscript and not elsewhere. Keep them the same for consistency.
Thank you very much for your comment. We have updated the manuscript as suggested by the reviewer.
- Reference 10 title should be corrected to the following: Projected prevalence and mortality associated with alcohol-related liver disease in the USA, 2019-40: a modelling study
Thank you very much for your comment. We have updated the reference as suggested by the reviewer

Yours Sincerely,

Dr Mohsan Subhani
MBBS, MRCP Medicine, MRCP Gastroenterology
Nottingham Digestive Diseases Biomedical Research Centre (NDDC)
University of Nottingham UK

References

1. Alcoholism NIAAA. Helping patients who drink too much: a clinician's guide, updated 2005 edition. Rockville: National Institutes of Health. 2005.
2. Organization WH. The ICD-10 Classification of Mental and Behavioural Disorders: Clinical descriptions and diagnostic guidelines. https://www.who.int/substance_abuse/terminology/ICD10ClinicalDiagnosis.pdf?ua=. Published 1990. Accessed.
3. Saunders JB, Degenhardt L, Reed GM, Poznyak V. Alcohol Use Disorders in ICD-11: Past, Present, and Future. *Alcoholism, clinical and experimental research*. 2019;43(8):1617-1631.
4. Subhani M, Knight H, Ryder S, Morling JR. Reply to Smith et al. Regarding 'Does Advice Based on Biomarkers of Liver Injury or Non-Invasive Tests of Liver Fibrosis Impact High-Risk Drinking Behaviour: A Systematic Review with Meta-Analysis'. *Alcohol and Alcoholism*. 2021;56(5):626-627.
5. Sim J, Lewis M. The size of a pilot study for a clinical trial should be calculated in relation to considerations of precision and efficiency. *Journal of clinical epidemiology*. 2012;65(3):301-308.
6. Lancaster GA, Dodd S, Williamson PR. Design and analysis of pilot studies: recommendations for good practice. *Journal of evaluation in clinical practice*. 2004;10(2):307-312.
7. Browne RH. On the use of a pilot sample for sample size determination. *Statistics in medicine*. 1995;14(17):1933-1940.
8. Sekhon M, Cartwright M, Francis JJ. Acceptability of healthcare interventions: an overview of reviews and development of a theoretical framework. *BMC Health Services Research*. 2017;17(1):88.
9. Braun V, Clarke V. Reflecting on reflexive thematic analysis. *Qualitative Research in Sport, Exercise and Health*. 2019;11(4):589-597.

VERSION 2 – REVIEW

REVIEWER	Thiele, Maja Odense Universitetshospital, Department for Gastroenterology and Hepatology
REVIEW RETURNED	30-Sep-2021

GENERAL COMMENTS	Thank you for the author reply letter and revised version of the study protocol. I only have two minor suggestions: - The strengths & limitations list seem somewhat over-optimistic, including only one limitation versus six strengths. Could the authors make the list more balanced? - Could the KLIFAD acronym be explained, for example in the abstract?
--

REVIEWER	Weerasinghe, Ashini Public Health Ontario, Health Promotion, Chronic Disease & Injury Prevention
REVIEW RETURNED	07-Oct-2021

GENERAL COMMENTS	The authors have made all the changes that I have previously asked for.
---

VERSION 2 – AUTHOR RESPONSE

Reviewer: 1

Comments to the Author:

Thank you for the author reply letter and revised version of the study protocol. I only have two minor suggestions:

- The strengths & limitations list seem somewhat over-optimistic, including only one limitation versus six strengths. Could the authors make the list more balanced?

Thank you very much for your comment. We have now merged the first two strengths and have added an additional limitation as suggested by the reviewer.

- Could the KLIFAD acronym be explained, for example in the abstract?

Thank you very much for your comment. We have now explained the KLIFAD acronym at the start of the methods section.

Reviewer: 2

Comments to the Author:

The authors have made all the changes that I have previously asked for.

Thank you very much.

Yours Sincerely,

Dr Mohsan Subhani

MBBS, MRCP Medicine, MRCP Gastroenterology

Nottingham Digestive Diseases Biomedical Research Centre (NDDC)

University of Nottingham UK